# Metabolic Insights into Caffeine’s Anti-Adipogenic Effects: An Exploration through Intestinal Microbiota Modulation in Obesity

**DOI:** 10.3390/ijms25031803

**Published:** 2024-02-02

**Authors:** Isabela Monique Fortunato, Quélita Cristina Pereira, Fabricio de Sousa Oliveira, Marisa Claudia Alvarez, Tanila Wood dos Santos, Marcelo Lima Ribeiro

**Affiliations:** 1Laboratory of Immunopharmacology and Molecular Biology, Sao Francisco University, Av. Sao Francisco de Assis, 218, Braganca Paulista 12916-900, SP, Brazil; fortunato.misabela@gmail.com (I.M.F.); quelitapereirapa@gmail.com (Q.C.P.); fabricio1445@hotmail.com (F.d.S.O.); marisacalvarez@yahoo.com (M.C.A.); tanilawood@gmail.com (T.W.d.S.); 2Hematology and Transfusion Medicine Center, University of Campinas/Hemocentro, UNICAMP, Rua Carlos Chagas 480, Campinas 13083-878, SP, Brazil

**Keywords:** obesity, gut microbiota, intestinal metabolites, metabolic insights, caffeine, anti-adipogenic effects

## Abstract

Obesity, a chronic condition marked by the excessive accumulation of adipose tissue, not only affects individual well-being but also significantly inflates healthcare costs. The physiological excess of fat manifests as triglyceride (TG) deposition within adipose tissue, with white adipose tissue (WAT) expansion via adipocyte hyperplasia being a key adipogenesis mechanism. As efforts intensify to address this global health crisis, understanding the complex interplay of contributing factors becomes critical for effective public health interventions and improved patient outcomes. In this context, gut microbiota-derived metabolites play an important role in orchestrating obesity modulation. Microbial lipopolysaccharides (LPS), secondary bile acids (BA), short-chain fatty acids (SCFAs), and trimethylamine (TMA) are the main intestinal metabolites in dyslipidemic states. Emerging evidence highlights the microbiota’s substantial role in influencing host metabolism and subsequent health outcomes, presenting new avenues for therapeutic strategies, including polyphenol-based manipulations of these microbial populations. Among various agents, caffeine emerges as a potent modulator of metabolic pathways, exhibiting anti-inflammatory, antioxidant, and obesity-mitigating properties. Notably, caffeine’s anti-adipogenic potential, attributed to the downregulation of key adipogenesis regulators, has been established. Recent findings further indicate that caffeine’s influence on obesity may be mediated through alterations in the gut microbiota and its metabolic byproducts. Therefore, the present review summarizes the anti-adipogenic effect of caffeine in modulating obesity through the intestinal microbiota and its metabolites.

## 1. Introduction

Obesity, like other chronic diseases, is accompanied by an inflammatory process generated by excessive fat accumulation when compared to predicted values for height, gender and age. Furthermore, it is considered a complex disease with multifaceted etiology, with its own pathophysiology, comorbidities, and disabling capabilities [1,2]. Excess physiologically optimal fat content allows triglyceride (TG) deposition by adipose tissue, enlarging the size of lipid droplets, expanding fat tissue, and resulting in obesity [3,4,5]. The expansion of white adipose tissue (WAT) is characterized by an increase in the size and number of adipocytes, respectively, called hypertrophy, which is linked to lipogenesis and hyperplasia, the mechanism of which is significant in adipogenesis [6].

Understanding the complex interplay of the contributing factors becomes critical for effective public health interventions and improved patient outcomes. Considering this, it is recognized that the intricate interaction between the microbiota and obesity goes beyond a mere correlation. Several studies indicate that the intestinal microbiota—in addition to being considered an endocrine organ involved in maintaining energy homeostasis and host immunity [7,8,9]—has been recognized as an important factor in the development of metabolic diseases such as obesity. Several studies have already associated a high-fat diet with a reduction mainly in Bacteroidetes and an increase in *Firmicutes*-to-*Bacteroidetes* ratio, among other bacteria, to a lesser extent [10,11,12,13,14,15].

Research evidence links gut microbiota-derived metabolites and obesity [16,17,18,19,20,21,22], recognizing that the identification of circulating metabolites from plasma, body fluids, and intracellular compartments contributes as a potential therapeutic target for interventions that seek to optimize metabolic and immunological results. Microbial lipopolysaccharides (LPS), secondary bile acids (BA), short-chain fatty acids (SCFAs), and trimethylamine (TMA) are the main intestinal metabolites in dyslipidemic states, followed by atherosclerotic cardiovascular disease (ASCVD) [19].

Previous studies have clearly demonstrated the anti-adipogenic effects of caffeine through the inhibition of key genes in adipogenesis, specifically PPARγ2 and C/EBPα [23,24]. In addition to its ability to modulate metabolism, with anti-inflammatory, antioxidant, and obesity-reducing effects [25,26], more recently, it was demonstrated that caffeine modulates obesity through the intestinal microbiota and its metabolites [27,28,29,30,31]. Therefore, the present review summarizes the anti-adipogenic effects of caffeine through modulation of the intestinal microbiota.

## 2. Physiological and Molecular Characterization of Obesity

Recognized as a chronic condition, obesity is characterized by the excessive accumulation of adipose [32]. Recent data from the World Health Organization (WHO) reveals a staggering increase in obesity prevalence, nearly tripling from 1975 to 2016 and impacting a substantial 650 million adults globally. Notably, being overweight has now touched 39% of the world’s population, highlighting the widespread nature of this health concern [33]. This significant surge in obesity rates underscores the mounting public health challenge it poses. Beyond the immediate impact on individual well-being, obesity contributes substantially to increased healthcare expenditures. Managing associated comorbidities, including type 2 diabetes mellitus, hypertension, ischemic events, acute myocardial infarction, and other coronary diseases, places a considerable economic burden on healthcare systems globally [34]. The intersection of lifestyle factors, environmental influences, and genetic predispositions further complicates the multifaceted nature of obesity. As efforts intensify to address this global health crisis, understanding the complex interplay of contributing factors becomes paramount for effective public health interventions and improved patient outcomes.

Nonetheless, adipose tissue assumes pivotal metabolic regulatory functions, including systemic metabolic homeostasis, fat reservoir maintenance, and thermoregulation, as well as the orchestration of lipid mobilization control [3,5]. However, exceeding the physiologically optimal fat content leads to its deposition in the form of triglycerides (TG) by adipose tissue through the lipogenic pathway. This adipocyte-based storage process is intricately linked to the amplification of lipid droplet size, adipose tissue expansion, and consequent obesity [3,4,5].

Adipose tissue has four types: WAT (White Adipose Tissue), BAT (Brown Adipose Tissue), BeAT (Beige Adipose Tissue), and PAT (Pink Adipose Tissue). WAT serves as a pivotal hub for energy storage, release, and adipokine secretion, notably leptin and adiponectin, which are crucial for energy homeostasis. Its regulatory role extends to energy balance through orchestrated lipogenesis and lipolysis processes [3,5]. Composed of adipocytes, fibroblasts, pre-adipocytes, and immune cells, WAT actively contributes to the establishment of a pro-inflammatory microenvironment during obesity. This phenomenon results from adipose tissue expansion, characterized by hypertrophy (the enlargement of adipocyte size) linked to lipogenesis and hyperplasia (an increase in adipocyte numbers), a significant mechanism in adipogenesis [5]. BAT is morphologically distinguished by numerous multilocular lipid droplets, abundant mitochondria, and elevated gene expression of brown adipocytes, particularly Uncoupling Protein 1 (UCP1). Augmentation in the brown adipocyte population correlates with the instigation of an anti-inflammatory phenotype, diminished insulin resistance, heightened thermogenesis, and consequent mitigation of obesity [35,36,37,38]. It has been reported that WAT exposure to stimuli, such as the cold, alterations in the microbiota, and activation of adrenergic receptors, induces morphological and functional changes [39]. These alterations enable WAT to acquire characteristics similar to BAT through a process known as ‘browning’ [40]. A discernible marker for this transition is the sustained elevation of UCP1. Originating within WAT, these beige adipocytes present analogous features to BAT, thereby constituting a distinct adipose subtype, acknowledged as BeAT [35,36]. Furthermore, beige adipocytes demonstrate a unique capability to dynamically switch between the phenotypes associated with energy storage and dissipation [38]. Lastly, PAT is the fourth type of adipocyte recently described in the literature. The specific color of these adipocytes is acquired during pregnancy and lactation, resulting in the reversible conversion of white adipocytes (located in the mammary glands) into milk-producing epithelial cells (pink adipocytes). These cells are considered alveolar structures, characterized by numerous cytoplasmic lipid droplets responsible for milk secretion [41]. 

Obesity initiates a multifaceted inflammatory cascade, which activates the innate immune system and generates deleterious results such as fibrosis and necrosis. This intricate process involves the release of a plethora of adipokines, orchestrating the modulation of immune cells, particularly macrophages, which assume either a polarized M2 state or an alternatively activated phenotype [42]. M2-polarized macrophages play a pivotal role by secreting anti-inflammatory cytokines, including Interleukin 10 (IL-10), thus preserving insulin sensitivity in adipocytes. Conversely, M1-polarized macrophages adopt a pro-inflammatory status, releasing cytokines such as Tumor Necrosis Factor Alpha (TNF-α). The ensuing recruitment of these macrophages amplifies both the total macrophage count and the M1-to-M2 ratio, which is indicative of obesity-induced adipose tissue inflammation. This intricate interplay is intimately associated with insulin resistance and the pathogenesis of metabolic disorders [42]. Moreover, the dysregulated immune response in obesity leads to the aberrant activation of adipose tissue-resident immune cells, further exacerbating the inflammatory milieu. The resulting imbalance in macrophage polarization and cytokine release allows chronic low-grade inflammation to perpetuate, increasing the risk of comorbidities such as type 2 diabetes and cardiovascular disease [5,42].

Free fatty acids (FFA) wield the capacity to incite inflammation through an indirect binding affinity with Toll-like receptors (TLR) 4 and TLR2, mediated by the protein Fetuin-A. This intricate process culminates in the activation of the NF-κB (Nuclear Factor Kappa B) and JNK (c-Jun N-terminal kinase) 1 pathways [43]. Subsequent to their activation, these pathways exhibit the capability to augment the synthesis and secretion of chemokines, including Monocyte Chemoattractant Protein (MCP) 1, which allows accelerated infiltration of pro-inflammatory macrophages. Notably, the interplay of FFAs with TLRs underscores a pivotal link between nutrient metabolism and innate immune responses. The resultant activation of NF-κB and JNK1 not only orchestrates a pro-inflammatory milieu but also contributes to the dysregulation of metabolic homeostasis. These molecular events further elucidate the intricate connections between lipid metabolism, inflammation, and the pathogenesis of obesity-associated complications, including insulin resistance and metabolic syndrome [4,5].

Other evidence suggests that hypoxia induces inflammation in obesity through adipocyte hypoperfusion or increased oxygen consumption [44]. The exposure of adipose tissue (AT) to hypoxic conditions in cultures demonstrates a positive regulation of numerous pro-inflammatory genes, correlating with regions exhibiting macrophage infiltration [45]. Consequently, infiltrated macrophages and adipocytes are responsible for the upregulation of inflammatory markers, such as TNF-α, IL-6, C-reactive protein (CRP), and MCP-1, both locally and in distant organs. These responses are stimulated by factors secreted in white adipose tissue [46]. The intricate network of cytokines, such as TNF-α, TNF-β, IL-1, IL-6, and Interferon (IFN), are deeply entwined with adipocyte metabolism, recognized for their nuanced roles as endogenous regulators of gene expression in adipose tissue. This regulatory network reverberates through various tissues, exerting profound effects on glucose homeostasis. Notably, TNF-α, with direct implications on adipocyte metabolism, exerts suppressive effects on the expression of numerous adipose tissue-specific genes. Its administration causes an increase in serum triglycerides and VLDL (very-low-density lipoprotein) in rats and humans. Long-term treatment of adipocytes with TNFα negatively regulates the expression of Glucose transporter type 4 (GLUT4), also denominate SLC2A, which indicates that TNFα may be a key mediator in abnormal gene expression in syndromes that correlate obesity and diabetes and that consequently affect glucose homeostasis [3,5,42]. 

## 3. The Impact of Obesity on the Microbiota

### 3.1. Intestinal Microbiota

The intestinal microbiota constitutes a complex assembly of approximately 100 trillion anaerobic bacteria [47]. Ongoing investigations elucidate that the intestinal microbiota is predominantly characterized by 80–90% Bacteroidetes (Gram-negative, exemplified by Bacteroides and Prevotella) and Firmicutes (Gram-positive, including Clostridium, Enterococcus, Lactobacillus, and Ruminococcus), alongside Actinobacteria (Gram-negative, e.g., Bifidobacteria) and Proteobacteria (Gram-negative, e.g., Helicobacter and Escherichia) [10,48,49]. The intricate phylum-level composition of the microbiota assumes a pivotal role in human metabolism by finely modulating the host’s nutritional dynamics and energy utilization [7,8,9]. 

The microbiota accomplishes this through the synthesis of essential vitamins, such as K, folic acid, and B12, along with the facilitation of electrolyte and mineral absorption. Furthermore, the microbiota actively participates in the fermentation of indigestible dietary components, contributing to metabolic processes within the host [8]. Beyond its metabolic contributions, the microbiota significantly influences the homeostasis of the intestinal epithelium and orchestrates intricate interactions with the host’s immune system [50,51,52]. This nuanced relationship underscores the microbiota’s pivotal role in shaping the host’s overall health and emphasizes its potential as a therapeutic target for interventions seeking to optimize metabolic and immunological outcomes [53].

The microbiota produces bioactive components from the metabolization of elements from the diet. Gut microbes metabolize indigestible carbohydrates, including cellulose, hemicelluloses, resistant starch, pectin, oligosaccharides, and lignin, into short-chain fatty acids (SCFAs) such as acetic, propionic, and butyric acids. SCFAs escape digestion through the gastrointestinal tract; they eventually enter the colon, and therefore, the exacerbated increase is associated with disease in the host [54,55,56]. Furthermore, the intestinal microbiota synthesizes gamma amino butyric acid (GABA), an important inhibitory neurotransmitter in the brain [57], carbohydrates, branched-chain amino acids, amines, phenols, indoles, and phenylacetic acid [58]. This is in addition to the relationship with the synthesis of bile acids, cholesterol, and conjugated fatty acids [59]. SCFAs, resulting from the microbial fermentation of indigestible polysaccharides, notably dietary fibers, act as crucial metabolites produced by the commensal microbiota [60,61]. These SCFAs, including acetate, propionate, and butyrate, are utilized as energy resources by both gut microbes and circulating molecules with systemic effects. Beyond the intestines, SCFAs influence various physiological processes. They contribute to hepatic gluconeogenesis, impacting glucose metabolism in the liver. Additionally, SCFAs are implicated in adipocyte lipogenesis, influencing the storage of fats in adipose tissue [62]. Their role extends to serving as signaling molecules by interacting with receptors throughout the body. The intricate interplay between microbial fermentation, SCFA production, and their systemic effects underscores the multifaceted impact of the gut microbiota (GM) on host metabolism and homeostasis [63,64,65,66]. 

SCFAs can promote lipogenesis and consequently increase triglycerides through the positive regulation of the expression of important genes in these processes, such as carbohydrate-responsive element-binding protein (CHREBP) and sterol regulatory element-binding transcription factor-1 (SREBP-1), and the downregulation of fasting-induced adipocyte factor (FIAF) [67]. Since SCFAs are in the bloodstream, their binding to GPCRs promotes the activation of lipid, glucose, and cholesterol metabolism [62] and inflammation in the intestine [68]. Evidence indicates that SCFAs linked to GPR41 (FFAR3) and GPR43 (FFAR2) receptors promote the increase in glucagon-like peptide-1 (GLP-1) in L cells [69,70]. It is known that the SCFAs present in greater quantities are acetate, propionate, and butyrate [71], and their role in several relevant signaling pathways is already recognized. Butyric acid performs activities responsible for suppressing the immune system and preserving the healthy balance of the intestine by stimulating the connection between IL-8 and GPR109A, in addition to inhibiting histone deacetylases (HDACs) and controlling the inflammatory process [72]. Additional studies demonstrate that propionate and butyrate generate soluble fiber fermentation products capable of stimulating gluconeogenesis in the intestine through specific genes involving the cAMP pathway or a relationship between the intestine and the brain [73]. Furthermore, acetate can precede acetyl-CoA and fatty acids used in de novo lipogenesis (DNL) in the liver. Its exacerbated increase may be involved in obesity and liver diseases related to non-alcoholic fats [74,75,76,77].

Commensalism is the term used to designate the type of relationship between the host and intestinal microbes, which allows maintenance [69] and maintains homeostatic balance. A feedback system is necessary, but if this is imbalanced, the number of microbes will consequently be altered because the cooperation of the microbial community has been disturbed. Considering a positive feedback system, a microbial community composed of two species will have an increase in abundance of both, which can generate an uncontrolled effect [78,79]. However, negative feedback results in the need for selection among cooperating microbes to maintain homeostasis in the gut. Therefore, this imbalance significantly disrupts the overall diversity and abundance of a community in an ecological niche [80]. Age, diet, genetics, medications, and disease development are factors that alter feedback mechanisms, resulting in changes in the composition of these microbes [81,82,83,84,85,86,87]. 

Colonization of the intestinal microbiota begins during pregnancy through commensal bacterial transmission from mother to child via the placenta [88,89]. Babies of 1 to 6 months have GM characterized by the colonization of *Bifidobacterium* and *Collinsella*, and only at 3 years of age is the complexity of the adult microbiome reached, although, at 12 months, there is already a similar microbiota [90,91]. 

In healthy adults, the microbial community is relatively stable, except in elderly people, which presents a more unstable and abundant community. As previously reported, the dominant flora that colonizes the intestine is composed of Firmicutes and Bacteroidetes, in addition to other species to a lesser extent. However, a healthy state is necessary for these bacteria to perform, among other functions, those related to metabolism, modification, methanogenesis, lipopolysaccharide biosynthesis, and oxidative phosphorylation [92,93]. 

The development of obesity and its related diseases is one of the factors that alter intestinal composition, resulting in an intestinal imbalance known as dysbiosis. Studies have already associated a high-fat diet with a reduction in Gram-negative organisms (including Bacteroides) and Gram-positive bacteria (including *Eubacterium rectale*, *Clostridium coccoides*, and *Bifidobacterium*) [10,11], in addition to increasing the *Firmicutes*-to-*Bacteroidetes* ratio, *Lactobacillus* spp., *E. coli*, *Prevotellaceae*, *Archaea*, and Firmicutes [10,12,13,14,15].

Notably, these microbial communities go beyond their conventional roles, extending their impact on the host’s overall health through intricate interactions with various physiological processes. Moreover, recent research emphasizes the microbiota’s ability to produce bioactive compounds and metabolites, influencing the host’s physiology and contributing to the regulation of systemic functions [94,95,96]. Zwartjes et al. (2021) delineated the pivotal role of key intestinal metabolites in dyslipidemic states followed by atherosclerotic cardiovascular disease (ASCVD), microbial lipopolysaccharides (LPS), secondary BA, SCFAs, and trimethylamine (TMA). This expanding knowledge underscores the profound implications of the microbiota in shaping the host’s health, with implications for therapeutic interventions aimed at manipulating these microbial communities for improved well-being.

### 3.2. Intestinal Microbiota and Obesity

It is now recognized that the intricate interplay between the microbiota and obesity extends beyond a mere correlation [97]. Specifically, it has been observed that genetically obese mice or those exhibiting diet-induced obesity reveal the microbiota as a potential causal factor in the accumulation of fat, a phenomenon decoupled from changes in calorie intake [98]. Groundbreaking research by Bäckhed and Ding et al., 2004, marked an early milestone in this exploration by comparing mice with a germ-free (GF) status—characterized by a sterile intestinal environment—to conventional mice and unveiled a substantial 42% increase in fat content in the latter under baseline conditions Adding depth to this narrative, [99] conducted studies that went beyond mere association. Colonizing germ-free rodents with microbiota that are derived from conventional counterparts caused a considerable increase in both weight and overall body mass. Intriguingly, this effect persisted independently of the alterations in total energy intake and expenditure [99]. These findings underscore the pivotal role of the microbiota in influencing metabolic dynamics and substantiate the notion that microbial communities play a contributory role in the complex landscape of obesity, offering avenues for deeper exploration and potential therapeutic interventions [100,101].

In light of this, it is comprehensible that the composition of the microbiota stands as a pivotal factor in triggering obesity. Notably, empirical evidence supports the notion that discerning signaling pathways modulated by the microbiota can elucidate the mechanisms behind heightened energy storage. The microbial colonization within the intestinal milieu has been instrumental in amplifying triglyceride (TG) storage through dual pathways. First, the processing of polysaccharides derived from the diet culminates in escalated hepatic lipogenesis, a phenomenon intricately linked to the augmented expression of CREBP/SREBP-1. Second, there is an elevation in the activity of Lipoprotein Lipase (LPL), achieved by the suppression of the FIAF within the intestinal epithelium [98].

Furthermore, additional studies have delved into the intricate dynamics linking microbial colonization in the intestine with the nuanced process of polysaccharide metabolism. Notably, the distal intestinal microbiota in humans has been documented to possess an extraordinary proficiency in acquiring and breaking down polysaccharides derived from plant sources. This phenomenon is complemented by the observed correlation between polysaccharide processing and the induction of monosaccharide transporters within the host [102,103]. In a murine model, a compelling observation emerged as C57BL/6J mice transplanted with fecal microbial communities from adult humans exhibited a discernible increase in adiposity [104]. This finding underscores the intricate interplay between microbial composition, polysaccharide metabolism, and host metabolic outcomes.

In light of the multifaceted nature of this discussion, some authors [78,105,106] have proposed that the intricate composition of GM is subject to dynamic variations over time, influenced by diverse factors such as age, genetic predispositions, environmental influences, and even among individuals adhering to identical dietary patterns. This variation is observed through an increase in the Firmicutes-to-Bacteroidetes ratio in obese people and is, therefore, an indication of obesity and cardiometabolic diseases. Extensive investigations across murine and human models have robustly substantiated this hypothesis [10,107,108,109]. However, recent evidence indicates that this reasoning should not be the sole basis for attributing obesity to the microbiota. It has been emphasized that it is necessary to incorporate factors such as the ecological composition of the intestinal microbiota, i.e., correlating microorganisms with their environment, along with the examination of produced metabolites, components, and functional changes contributing to the observed dyslipidemia [110,111].

## 4. Metabolites with the Potential to Reshape the Microbiota in the Context of Obesity

Evidence from research studies unequivocally demonstrates the profound influence of the metabolites derived from the intestinal microbiota in orchestrating obesity modulation. Recognizing this intricate relationship necessitates the application of metabolomics—a comprehensive approach delving into the identification of metabolites not only circulating in plasma but also within bodily fluids and intracellular compartments. These metabolites, originating from the complex milieu of GM, exhibit discernible correlations with the intricate landscape of obesity [19,96,112,113,114]. 

Intestinal modulation extends to hypersecretion of very-low-density lipoprotein (VLDL), the induction of non-alcoholic fatty liver disease (NAFLD), the blockade of reverse cholesterol transport, and the potential induction of adipocyte disruption or inflammation [19]. Further research supports the association of LPL dysfunction as a primary pathogenic mechanism in dyslipidemia among obese individuals [98,115,116]. This linkage is attributed to the influence of GM and its metabolites on proteins that intricately regulate LPL-mediated lipolysis. Among these proteins are adipocytes and signaling molecules, including Angiopoietin-like Proteins (ANGPTLs) 3 and, notably, ANGPTL4—formerly recognized as Fasting-Induced Adipose Factor (FIAF). Additionally, apolipoproteins like Apo-A5 contribute to this intricate regulatory network [19].

Diving into the intricacies of ANGPTL4’s functionality, it becomes evident that its inhibitory influence on LPL activity disrupts the hydrolysis of TG, impeding the efficient transfer of FFA to adipocytes and other tissues. The complex interplay between metabolites originating from GM and LPL functions is underscored by the modulation of protein secretion. An in-depth exploration of scientific findings reveals that an elevation in ANGPTL4 levels leads to an increased presence of TG-rich lipoproteins, notably VLDL [20,21], and chylomicrons, contributing significantly to adipocyte storage [117,118,119,120,121]. Turning attention to Apo-A5, a pivotal player in reducing the TG levels in plasma by enhancing LPL activity, investigations highlight a nuanced correlation between the abundance of specific microbial species (such as the Porphyromonadaceae family, including the Parabacteroides genera, the Rikenellaceae family, and Odoribacter genus) and Apo-A5 polymorphisms, which are intricately associated with the manifestation of hypertriglyceridemia and reductions in Actinobacteria such as Bifidobacterium [22]. Recent evidence strongly suggests a correlation between the observed microbial composition and the development of NAFLD. NAFLD is recognized by the exaggerated increase in VLDL, which is considered the crucial lipoprotein regulated, in part, by Apo-A5. This regulatory process may be influenced by various metabolites originating from GM, including microbial LPS, TMA and its oxidized form trimethylamine N-oxide (TMAO), secondary BA, and by interaction with other ligands, as Farnesoid X Receptor (FXR) [122,123,124].

TMA originates from the diet and is metabolized through L-carnitine, choline, or phosphatidylcholine. Thus, it is considered one of the metabolites of the intestinal microbiota. It can subsequently be converted into TMAO in the liver, which represents its oxidized form [125]. Moreover, elevated levels of TMAO have been implicated in a reduction in cholesterol efflux from tissues to the liver. This effect is associated with a decrease in the expression of bile acid transporters, leading to a compromised elimination of cholesterol. The intricate relationship between increased TMAO and compromised Reverse Cholesterol Transport (RCT) is well-documented [126]. Furthermore, TMAO has been implicated in increasing the risk of cardiovascular disease (CVD) [60]. It is noteworthy that the activity of RCT is highly dependent on High-Density Lipoprotein Cholesterol (HDL-C), which is responsible for facilitating the efflux of cholesterol from tissues to the liver. The regulatory role is performed by the Liver X Receptor (LXR) α or β [127]. Therefore, the observed increase in TMAO levels may impact this essential mechanism, contributing to reverse disorders related to cholesterol homeostasis and metabolic health [126,128].

It has been demonstrated that the Influence of GM through TLR4, activated by LPS derived from Gram-negative bacteria, and TLR-5, activated by viral RNA, can inhibit LXR. The inhibition of LXR subsequently impacts HDL-C, thereby disrupting the efflux of cholesterol [125]. Another layer of complexity emerges when considering the role of LXR and Flavin monooxygenase isoform 3 (FMO3) in the conversion of TMA to TMAO. Inactivation of FMO3 has been associated with an increased cholesterol efflux mediated by LXR [128]. TLR-5, despite its role in inflammation by recognizing flagellin as a Pathogen-Associated Molecular Pattern (PAMP), also exhibits a dual nature [129]. Yiu et al. (2020) demonstrated that TLR-5 induces the formation of Apolipoprotein A-I (Apo-AI), a critical early step in HDL synthesis. These intricate molecular interactions highlight the multifaceted involvement of GM in lipid metabolism and inflammatory pathways, shedding light on the intricate mechanisms underlying metabolic disorders [130].

TMAO’s pro-atherogenic effects, as evidenced in numerous studies, stem from its role in increasing dyslipidemia and elevating the risk of cardiovascular diseases. The underlying mechanism involves TMAO influencing the extent of cholesterol deposition in peripheral tissues, particularly within macrophages located in the arterial intima. This influence occurs through the regulation of receptors A1 and CD36. These receptors play crucial roles in gathering oxidized low-density lipoprotein (LDL) and lipid-laden macrophages, contributing to the progression of cardiovascular diseases [125,126,131,132,133,134].

As mentioned previously, the FXR, a bile acid receptor originating from GM, and the G protein-coupled bile acid receptor 1 (GPBAR-1 or TGR-5) share a microbial origin. However, FXR, in contrast to TGR-5, is intricately linked to lipid metabolism, as substantiated by several studies [135,136,137,138]. The inactivation of this gene has been correlated with hypertriglyceridemia and elevated total cholesterol levels. Conversely, its activation led to a decrease in triglycerides by downregulating Sterol Regulatory Element-Binding Protein 1c (SREBP-1c), a de novo lipogenesis-related transcription factor. Furthermore, FXR activation enables the release of FFA by activating the PPARα pathway and inducing HDL synthesis [139,140]. The GPBAR-1 receptor operates indirectly in lipid metabolism but plays a crucial role in regulating circulating bile acids. Its activation has been linked to a reduction in body weight in murine models subjected to a high-fat diet. While GPBAR-1’s influence on lipid metabolism is not as direct as some other receptors, its impact on bile acid circulation and the potential downstream effects on metabolic processes make it a significant player in the complex network of metabolic regulation [141].

Finally, it has been suggested that SCFAs play a role in the conversion of cholesterol to its reduced form, known as coprostanol, rendering it non-absorbable and facilitating its elimination through feces. This conversion process is facilitated by microbial enzymes, and an increase in coprostanol levels results in a decrease in cholesterol levels [142]. SCFA significantly impacts cholesterol metabolism and influences important metabolic pathways, with potential implications for cardiovascular health [16,17,18]. 

In summary, metabolites are crucially important for GM remodeling, influencing obesity and metabolic processes. Figure 1 highlights the influence of the intestinal microbiota and its metabolites on organs and tissues, especially in the context of dyslipidemia. Understanding and manipulating these interactions could offer promising avenues for addressing obesity and related metabolic disorders. Among the strategies currently explored, caffeine stands out as an intriguing pathway, potentially impacting the metabolism and intricate dynamics of GMOs. Understanding these relationships helps unravel the complexities of obesity and metabolic health.

## 5. Caffeine—Mechanisms of Action on Obesity

Caffeine (1,3,7-trimethylxanthine) is a methylxanthine compound present in chocolate, coffee, and tea, widely consumed by humans globally [143]. Caffeine is recognized as a central nervous system stimulant, exerting its effects by blocking adenosine receptors A1 and A2 in the brain, owing to the structural resemblance between caffeine and adenosine. This interaction influences the release of neurotransmitters, including acetylcholine, dopamine, and norepinephrine, thereby modulating mood, enhancing alertness, and reducing physical fatigue [144,145,146,147,148].

Moreover, it has been shown that caffeine also possesses effects on oxidative metabolism and modulation of the inflammatory state in the central nervous system, preventing neurodegenerative diseases, including Alzheimer’s and Parkinson’s [149,150,151]. Caffeine supplementation has been shown to improve performance and endurance in physical exercise [152,153]. Caffeine has also demonstrated bronchodilator effects and reduced fatigue in respiratory muscles, associated with its anti-inflammatory action [154,155]. Furthermore, several studies point to caffeine’s ability to modulate metabolism, with anti-inflammatory and antioxidant effects, and a reduction in obesity [25,26,156]. 

Fatty acids and glycerol are obtained after the breakdown of triglyceride, the process of which is called lipolysis. Lipoprotein lipases, responsible for degradation, are activated by catecholamines (noradrenaline and adrenaline) and hormones (insulin, glucagon, and adrenocorticotropin). On the other hand, they can be inhibited by insulin because when it binds to the alpha-adrenergic receptor, it can promote fat retention, whereas beta, when it binds to a catecholamine or a hormone, stimulates fat to be manipulated [157]. 

Furthermore, conformational changes in the G protein-coupled receptor and the synthesis of cytosolic cAMP through the stimulation of adenylate cyclase are factors that are also linked to the activation of lipolysis. Cytosolic cAMP activates hormone-sensitive lipase (HSL) in its phosphorylated form after stimulating protein kinase A. This phosphorylated enzyme inhibits phosphodiesterases (PDE) capable of degrading cAMP into non-cyclic AMP. The reduction in cAMP is linked to the inhibition of lipolysis [158,159].

Positively, studies indicate that caffeine can modulate signaling pathways, which have been previously reported. This is because the compound can produce catecholamines through an increase in cAMP, which stimulates hormone-sensitive lipase (HSL) [160]. Another mechanism by which caffeine works is by inhibiting adrenergic receptors, which reduces the accumulation of fat and promotes the breakdown of triglycerides [161,162]. The compound is also capable of increasing cAMP by blocking PDE, thus promoting lipolysis [163].

Moreover, other studies have linked caffeine to β-oxidation activity. Helal et al. demonstrated that even under conditions of a high-fat diet, caffeine suppressed the expression of FAS, which is involved in hepatic lipid metabolism. This suppression promoted de novo lipogenesis and lipid β-oxidation through the activation of PPARα [164]. Furthermore, it is known that the upregulation of PPARα through FXR stimulates free fatty acids β-oxidation and, consequently, a reduction in VLDL [165]. The decrease in triglycerides associated with the upregulation of O-acylcarnitine suggests an increase in FFA β-oxidation and a consequent improvement in the lipid profile and related metabolic syndromes [166]. 

Furthermore, caffeine stimulated the BAT function through an increased expression of the peroxisome proliferator-activated receptor gamma 1-alpha coactivator (PGC1α), which promoted tissue thermogenesis and mitochondrial biogenesis. This is associated with other BAT-selective and beige gene markers [167,168]. Figure 2 represents a scheme that summarizes the mechanisms by which caffeine can act to promote lipolysis, β-oxidation, and thermogenesis.

Caffeine also exhibits anti-adipogenic properties by suppressing the differentiation of 3T3-L1 adipocytes. It inhibits the expression of master regulators in the adipogenic process, specifically PPARγ2 and C/EBPα [23,24]. Additionally, caffeine interferes with adipogenesis by inactivating the AKT/GSKβ (Glycogen Synthase Kinase Beta) pathway. The AKT pathway is crucial for adipocyte differentiation, and its involvement in the insulin or IGF1 signaling pathway is mediated, in part, by the phosphorylation of the GSK3β protein. Inactivation of the AKT pathway may also result in the deactivation of FKHR, a key element in the progression of adipogenesis [24]. 

Mitani et al. (2017) demonstrated that caffeine reduced lipid levels by stimulating Caco-2 cells, which come from the intestine and are responsible for regulating the secretion of inflammatory cytokines. 3T3-L1 adipocytes incubated with caffeine showed a decreased lipid abundance and suppression of genes, such as proliferator-activated receptor (PPAR) γ and CCAAT/C/EBP α in 3T3-L1 adipocytes. Furthermore, it reduced the expression of C/EBPβ and C/EBPδ at the protein level, although not at the mRNA level. It also suppressed interleukin-8 and plasminogen activator inhibitor-1 from Caco-2 [23].

Evidence associates caffeine with increased de novo lipogenesis through ameliorated hepatic steatosis and inflammatory injury in vivo, where the caffeine repressed AKT/mTORC1 and SREBP-1/ACC/FASN signaling in mice and in vitro. Furthermore, caffeine impaired NF-κB activation by stabilizing IκBα, resulting in a reduction in pro-inflammatory mediators interleukin-6 (IL-6) and tumor necrosis factor α (TNF-α) [169]. 

Comparing high-fat diet (HFD) vehicle rats (HFD-V) and HFD-caffeine rats, the inflammatory profiles of circulating immune cells were suppressed (TNFa, MCP-1, IL-6, intercellular adhesion molecule 1 (ICAM-1), and nitrite) and accompanied by decreased liver, white adipose tissue (WAT), and muscle macrophages and their intracellular cytokine levels. In addition to reducing lipid accumulation in various tissues, caffeine upregulated tissue lipogenic markers SREBP-1c, fatty acid synthase, and acetyl-CoA carboxylase)/insulin sensitizers (GLUT4 and p-IRS1) in HFD-caffeine rats [170].

In vitro, human HepG2 cells were used to evaluate the effect of caffeine, and the results demonstrated a relevant decrease in hepatic lipid accumulation, including triglycerides (TG) and cholesterol, in addition to decreasing the mRNA level of SREBP1c, SREBP2, Fatty acid synthase (FAS), Stearoyl-CoA desaturase 1 (SCD1), Hydroxymethyl glutaryl CoA reductase (HMGR), and low-density lipoprotein receptor (LDLR)—genes associated with lipogenesis. However, conversely, the level of CD36 mRNA increased; this receptor is involved in lipid acquisition and catabolism. Furthermore, phosphorylation of AMPK and acetyl-CoA carboxylase increased after 24 h with caffeine [171].

Caffeine improved the damage caused to the liver by HFD; this was observed through a reduction in the amount of alanine aminotransferase (ALT), aspartate aminotransferase (AST), and bilirubin present in the blood, in addition to providing an increase in albumin. Caffeine also decreased the level of mRNA for fatty acids and acetyl-CoA carboxylase, as well as the protein expression of carnitine palmitoyltransferase 1, and increased the expression of PPARα, and finally, it also decreased the oxidative degradation of lipids [164]. Additional studies have shown that caffeine intervention promoted a potent effect in reducing hepatic lipid levels through the activation of autophagy in cell cultures and in vivo. This mechanism allowed the inhibition of mTOR signaling, mobilization, and hydrolysis of TGs into FFA through the autophagy-lysosomal pathway, resulting in an increase in FFAs to mitochondria and, consequently, an increase in the flow of acylcarnitine, b-oxidation of fatty acids, and oxidative phosphorylation [172].

## 6. Impacts of Caffeine on Shaping Obesity through Intestinal Microbiota and Its Metabolites

Recently, it has been demonstrated that caffeine modulates obesity through intestinal microbiota and its metabolites. Caffeine promotes the degradation of lipids through the stimulation of catecholamines and an increase in the adrenergic receptors responsible for activating this process, as well as disrupting adipogenesis by inhibiting glucose transport in adipocytes [173]. In addition to inhibiting insulin-stimulated activity and adipogenesis in rodent adipocytes, other evidence demonstrated that caffeine inhibited the oxidized form of phenylmethylamine, compromising the production of primary amine oxidase substrates [174].

Despite having heart disease, interventions with caffeine improved both metabolic syndrome and associated lipid disorders and increased insulin sensitivity. The technique used was the sequencing of 16S rRNA in the microbiome, which ensures observation of how the compound affects the composition of the microbiome [175]. It relatively increased the levels of Dubosiella, Bifidobacterium, and Desulfovibrio while causing a decrease in Bacteroides, Lactobacillus, and Lactococcus. Furthermore, caffeine altered serum metabolomics, mainly affecting lipids, bile acids, and energy metabolism. Additionally, it increased 1,7-Dimethylxanthine, a metabolite related to Dubosiella [31]. 

The intestinal microbiota and serum metabolism were modulated after the consumption of caffeinated coffee by rats fed with HD because the compound can reduce the animals’ weight, adiposity, triglyceride level, and energy intake. Consumption also attenuated an increase in the proportion of Firmicutes to Bacteroidetes and Clostridium Cluster XI. The intestinal microbiota and serum metabolism were modulated after the consumption of caffeinated coffee by HFD-fed rats, as there was a decrease in body weight, adiposity, liver triglycerides, and energy intake. Consumption also attenuated an increase in the proportion of Firmicutes to Bacteroidetes and Clostridium Cluster XI. Furthermore, coffee resulted in a more favorable metabolomic profile, whether with the standard diet or HFD, evidenced by increased metabolites indicative of carbohydrate and fatty acid metabolism, including citrate, carnitine, pyruvate, and glycerol. Coffee also reduced the levels of BCAAs leucine and valine in both diets and reduced isoleucine in HFD—it is known that these metabolites are indicative of insulin resistance in obese people—and finally, in relation to SCFAs, the treatment increased the levels of acetate and propionate [29].

Lynch et al. (2023) correlated caffeine intake or altered metabolism in obese elderly individuals with a decrease of 5% or more in body weight, classified as a responder group [28]. They observed an increase in phytochemical metabolites and microbiome-related metabolites, mainly in responders. Many of these metabolites are part of the pipecolic acid metabolism, produced by the intestinal lysine microbiota, as suggested by [176]. Additionally, only responders demonstrated an increase in two indicators of the phenolic compound metabolism and one amino acid derivative. Finally, there was a significant increase in dimethylglycine, a metabolite of the vitamin-like compound choline [28].

In other association studies with caffeine, the compound also demonstrated similar effects. The Fubrick tea aqueous extract, which includes caffeine (4.6 mg), reversed HFD-induced GM dysbiosis by increasing the relative abundance of Bacteroidetes and reducing Staphylococcus. Furthermore, tea supplementation also modified the serum metabolome, mainly the “caffeine metabolism” pathway, through increased concentrations of caffeine, theophylline, and theobromine in the serum, which were positively correlated with the abundance of norank_f_Lachnospiraceae, whose relationship has previously been linked to anti-obesity effects [30].

Caffeine combined with Epigallocatechin-3-gallate (EGCG) has anti-obesity effects through increased beta diversity, decreased Firmicutes levels, and increased bifidobacterium, associated with increased fecal acetic acid and propionic acid [27] (both promoters of insulin sensitivity and satiety [177], and total SCFAs, which are related to the maintenance of mucosal integrity, expansion of gut microbes, and activation between the intestine, liver, and brain to obtain energy [178], as well as the decreased expression of the G-protein-coupled receptor (GPR43) 43, whose receptor is associated with an obese phenotype [179].

Furthermore, an increase in fecal loss of bAs and an increase in the copies of enzymes Bile Salt Hydrolases (BSH) in the intestine, responsible for hydrolyzing conjugated bAs into unconjugated BAs, was seen in intestinal bacteria. Unconjugated BAs are a prerequisite of 7α/β-dehydroxylation reactions for the production of secondary BAs. There is evidence already demonstrating that increasing BSH activity could improve BA deconjugation and subsequently increase BA fecal loss, improving obesity [180]. On the opposite hand, Chen et al. (2023) demonstrated that the decrease in BSH increased BA conjugates, consequently increasing the conversion of cholesterol into bile acids and reducing the accumulation of lipids. Concomitantly, at the plasma level, cholic acid decreased, and glycocholic acid increased after the intervention with caffeine [31].

Furthermore, Zhu et al. (2022) reported synergistic effects in relation to increased GPBAR-1 and a decreased expression of intestinal FXR–FGF15 (fibroblast growth factor 15), resulting in the increased expression of hepatic CYP7A1 [27], which plays a critical role in the catabolism of cholesterol and consequently the formation of bile acids in feces [181,182].

The treatment prevented the animals from gaining weight and inhibited lipid accumulation. It modulated the proportion of intestinal microbes, such as Lactobacillus, Bifidobacterium, Rikenella, Turicibacter, and Allobaculum, promoting the effects on metabolism. This included an increase in the amounts of theobromine, cysteic acid, L-cysteine, and 5′-methylthioadenosine, and a decrease in 3-hydroxyanthranilic acid and L-kynurenine, whose metabolites originate from tryptophan. Furthermore, the compound acted on the metabolism of biotin, resulting in its decrease, as well as increased chenodeoxycholic acid from the primary bile acids, and finally, decreased 7-dehydrodesmosterol, which is obtained through steroids [183]. Table 1 represents the main research related to the effects of caffeine in modulating obesity through the intestinal microbiota and its metabolites.

Post-fermented Pu-erh tea (PE) contains caffeine as one of the main chemical constituents, along with other polyphenols such as theabrownins, polysaccharides, and gallic acid (GA) [184,185], which can modulate the effects caused by tea. Song et al. demonstrated that polyphenols and caffeine, together with PE, promoted improvements in several metabolic parameters in animals fed a high-fat diet (HF), in addition to re-establishing intestinal homeostasis. These results are associated with changes in the intestinal microbiota, especially *Akkermansia muciniphila* and *Faecalibacterium prausnitzii* [186].

Other studies point to combinations between compounds, such as the green tea extract (GTE), which contains 5.1% caffeine, in association with EGCG and/or catechin (CAT). These combinations demonstrate altered metabolites that contribute to changes in metabolic profiles in these groups, likely impacting the host’s metabolic health through different biological pathways. These pathways include the regulated metabolism of NAD+ and other central energy metabolism (such as TCA and glycolysis), as well as the regulation of metabolites such as N-acetylserotonin, which has antioxidant activity. These combinations play an important role in the intestine with anti-obesity potential, modulating metabolic regulation independently of their impact on the microbial population [187].

The consumption of coffee and its main constituents, caffeine and chlorogenic acid, have been shown to partially repair the plasma SCFA profile, moderately modulating the intestinal microbiota, especially the Blautia, Coprococcus, and Prevotella bacteria, related to obesity, which suggested enabling the prevention of steatohepatitis in a model of metabolic syndrome [188].

In summary, caffeine significantly influences obesity dynamics through interactions with the gut microbiota and associated metabolites. Its mechanisms include promoting lipid degradation, inhibiting adipogenesis, and inducing alterations in serum metabolites. Across diverse models, caffeine consistently shows positive effects on weight regulation, adiposity reduction, and metabolic improvements. Combining caffeine with compounds like EGCG or within tea extracts demonstrates synergistic anti-obesity effects. These combinations affect microbial diversity, bile acid metabolism, and various metabolic pathways, leading to increased insulin sensitivity and satiety. Studies on specific populations, such as the elderly and responder groups, highlight individualized responses to caffeine, emphasizing associations with specific metabolites and microbiome-related compounds. Additionally, caffeine influences bile acid metabolism, hepatic enzymes, and plasma metabolites, offering potential therapeutic implications for managing obesity and metabolic disorders. In conclusion, caffeine’s direct impact on the gut microbiota and metabolites presents promising avenues for research and targeted interventions in the complex realm of obesity and metabolic syndrome. Continued investigation is crucial to understanding the intricate interactions and individual responses within diverse populations.

## 7. Conclusions

The current evidence highlights the relevance and complexity of obesity and related diseases. This review shows an amount of unprecedented data pointing to caffeine as a therapeutic potential in modulating obesity through its anti-adipogenic effect on the intestinal microbiota and its metabolites. In general, the data reviewed here clearly demonstrated that the intervention with caffeine inhibited adipogenesis, re-established the composition of the intestinal microbiota and the metabolic profile, and was associated with an improvement in the lipid profile and a reduction in body weight. However, the study of possible metabolites circulating in plasma, body fluids, and intracellular compartments is still a field that needs to be widely studied; therefore, it is urgently necessary to develop strategies capable of identifying the range of metabolites related to the modulation of obesity.

## Figures and Tables

**Figure 1 ijms-25-01803-f001:**
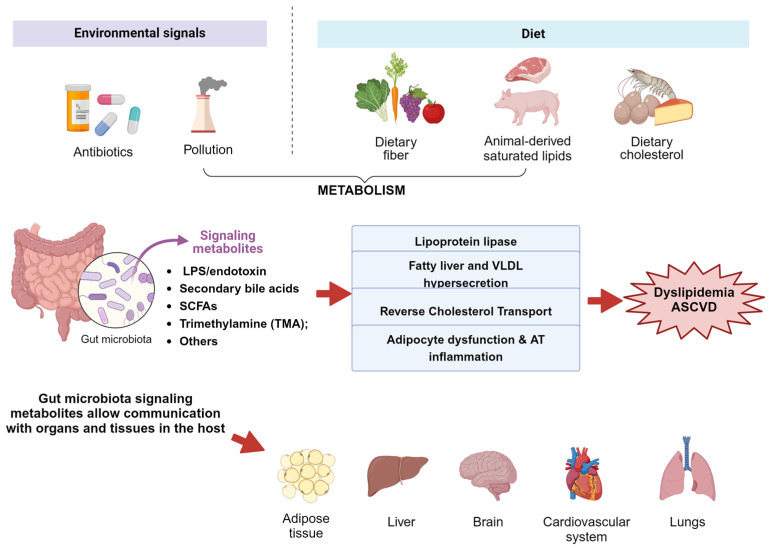
Influence of the intestinal microbiota and its metabolites on organs and tissues, especially in the context of dyslipidemia. The microbiota produces bioactive components through the metabolization of dietary elements, the process of which is influenced by environmental signals and the type of diet. Gut microbiota signaling metabolites enable communication with host organs and tissues. Metabolites produced by GMOs are linked to changes in the LPL function, NAFLD, inhibition of RCT, and adipocyte dysfunction or inflammation. Adapted from [19].

**Figure 2 ijms-25-01803-f002:**
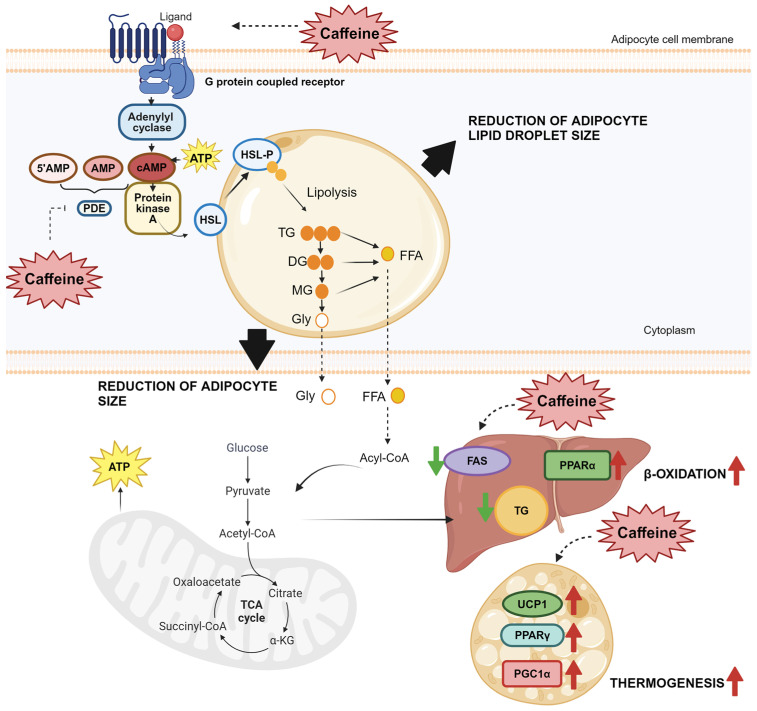
Caffeine exerts its mechanism of action during lipolysis in adipocytes, leading to the promotion of beta-oxidation and thermogenesis. Caffeine achieves a reduction in the size of adipocyte lipid droplets and adipocyte size through various mechanisms, such as an increase in catecholamine secretion and subsequent activation of hormone-sensitive lipase (HSL), blockade of adrenergic receptors, and elevation of cAMP levels. Additionally, caffeine promotes beta-oxidation by suppressing fatty acid synthase (FAS) and upregulating peroxisome proliferator-activated receptor alpha (PPARα). It also stimulates brown adipose tissue (BAT) by increasing peroxisome proliferator-activated receptor gamma coactivator 1-alpha (PGC1α), thereby promoting thermogenesis. Adapted from [157]. ↑ Increase.

**Table 1 ijms-25-01803-t001:** Effects of caffeine in modulating obesity through the intestinal microbiota and its metabolites.

Study Design	Key Finding	Ref.
3T3-L1 cells were pretreated with caffeine at concentrations of 0.25, 0.5, and 1.0 mM. Subsequently, they were stimulated with 100 nM insulin for 5 or 20 min.	Suppressed intracellular lipid accumulation after complete differentiation in a dose-dependent manner (0.125–1.0 mM). Inhibited insulin-stimulated glucose uptake.	[173]
The inhibitory effect of caffeine (0.1 and 1 mM) on lipogenesis in fat cells from wild-type mice and in mice with a mutation on the catalytic site of PrAO.	Inhibited basal and insulin-stimulated glucose transport. Inhibited lipogenesis in rodent adipose cells and adipocytes from mice genetically invalidated for PrAO activity. Inhibitory properties extended to human adipocytes.	[174]
C57BL/6 J mice were subjected to a 12-week treatment involving a normal chow diet or a high-fat diet with varying concentrations (0.05 or 0.1%) of caffeine. The assessment included parameters such as body weight, insulin resistance, lipid levels, GM, and metabolomics.	↑ Dubosiella, Bifidobacterium, and Desulfovibrio. ↓ Bacteroides; Lactobacillus Lactococcus Serum metabolomics alterated: lipid, bile acid, and energy metabolism; ↑ 1,7-Dimethylxanthine, positively correlated with Dubosiella; ↓ BSH; ↑conversion of cholesterol into bile acids; ↓ lipids accumulation; ↓ cholic acid in plasm level; ↑ glycocholic acid in plasm level.	[31]
Mice were randomly divided into a normal diet group, a high-fat diet group, and a high-fat plus Fuzhuan brick tea group and treated for 10 weeks.	↑ Bacteroidetes; ↓ Staphylococcus; ↑ Serum concentrations of caffeine, theophylline, and theobromine, which were positively correlated with the abundance of norank_f_Lachnospiraceae, whose relationship has previously been linked to anti-obesity effects.	[30]
Male Sprague–Dawley rats were divided into chow-water, chow-coffee, high-fat diet-water, and high-fat-coffee diets and treated for 10 weeks.	↑ ratio of Firmicutes to Bacteroidetes and Clostridium Cluster XI. ↑ metabolites, indicative of carbohydrate and fatty acid metabolism: citrate, carnitine, pyruvate, and glycerol; ↓ BCAA: leucine and valine; ↓ isoleucine and insulin resistance indicators; ↑ acetate and propionate.	[29]
The combined use of EGCG and caffeine was evaluated in rats after 6 weeks of treatment in the following groups: normal diet and high-fat diet.	↑ Beta diversity; ↓ Firmicutes level; and ↑ Bifidobacterium level. ↑ Fecal acetic acid; ↑ Propionic acid; ↑ Total SCFAs; ↓ GPR43; ↑ Fecal loss of Bas; ↑ BSH copies; ↑ GPBAR1; ↓ Intestinal FXR–FGF15; ↑ Hepatic CYP7A1.	[27]
Male C57BL-6J mice were fed with a normal diet and HFD, with 46 mg/mL extracted corn silk (caffeine content 10.46%).	↑ Plasma SCFAs; ↑ Lactobacillus, Bifidobacterium, Rikenella, Turicibacter, and Allobaculum. ↑ Theobromine levels in relation to caffeine metabolism; ↑ Cysteine and methionine metabolism; ↑ Chenodeoxycholic acid in primary bile acid biosynthesis; ↓ 7-dehydrodesmosterol in steroid biosynthesis. Catecholamines released ↑ adipocyte responses to adrenergic receptor-mediated lipolysis activation, glucose transport inhibited in adipocytes, insulin-stimulated activity inhibited, adipogenesis inhibited, oxidation of pheylmethylamine via primary amine oxidase inhibited, and primary amine oxidase substrates blocked.	[183]
Pilot pre/post-26-week technology-based weight loss intervention, which included exercise and nutrition components. It included 53 older adults (>65 years).	↑ Phytochemical metabolites and metabolites are related to the microbiome, and these various metabolites are part of the metabolism of pipecolic acid. ↑ Indicators of phenolic compounds metabolism, in addition to ↑ of an amino acid derivative. ↑ Dimethylglycine, a metabolite of the vitamin-like compound choline.	[28]

↑—induction; ↓—repression; HFD group = high-fat diet group; FTEs = Fubrick tea aqueous extract; EGCG = epigallocatechin-3-gallate; LE = low-dose EGCG; LC = low-dose caffeine; LE + LC = in combination; HE = high-dose EGCG; FXR–FGF15 = fibroblast growth factor 15; BCAA = branched-chain amino acids; CPC = co-fermented Pu-erh tea with an aqueous corn silk extract.

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
