# Peer review of "Metabolic Insights into Caffeine’s Anti-Adipogenic Effects: An Exploration through Intestinal Microbiota Modulation in Obesity"

_ijms, 2024, doi:10.3390/ijms25031803_

Round 1

Reviewer 1 Report

Comments and Suggestions for Authors

Reviewer’s comments

This manuscript by Fortunato et al., is an interesting scientific compilation. The article presents a balanced and critical perspective on the current evidence and knowledge gaps regarding caffeine’s anti-adipogenic effects. The authors are careful to acknowledge the limitations and challenges of the existing studies and suggest directions for future research. The manuscript provides a comprehensive review of the metabolic insights into caffeine’s anti-adipogenic effects, focusing on the role of intestinal microbiota and its metabolites in obesity modulation. There are several positives to be said about this manuscript. Which include 1) the article is well-structured, with clear headings, subheadings, tables, and figures. 2)The introduction and conclusion sections are concise and informative, summarizing the main points and implications of the review. 3) The article is well-referenced, with relevant and up-to-date sources from various journals and databases.

Minor modifications needed but not compulsory.

1)      The article could be improved by providing more details on the mechanisms and pathways involved in caffeine’s modulation of the intestinal microbiota and its metabolites, especially using figures to illustrate such mechanisms will greatly improve the manuscript.

2)      The authors could point out potential interactions and synergies with other dietary components or factors.

3)      The article could also discuss the clinical implications and applications of caffeine’s anti-adipogenic effects for obesity prevention and treatment.

Minor modifications needed, compulsory.

1)      Line 18. Space needed between the words “outcomes” and “In” in the phrase “outcomes.In”.

2)      Figure 1. Requires revision to enhance clarity. Resize the figure and improve the quality. What use is the big white arrow for, and why is it not mentioned in the body of the work? It will be good to commit to one type of arrow, unless the different arrow type is to typify a specific mechanism different from the general information been carried out.

3)      I prefer that a figure should be created to illustrate the caffeine mechanism of action on obesity. This will easily improve the manuscript and enhance the readership of this manuscript.

Author Response

I hope this letter finds you well. I appreciate the time and effort that you and the reviewers have dedicated to the review of our manuscript titled “Metabolic Insights into Caffeine's Anti-adipogenic effects: An exploration through Intestinal Microbiota Modulation in Obesity." We are grateful for the insightful comments and suggestions provided by the referees. We have carefully considered each of the points raised, and I would like to address them in detail.

Minor modifications needed but not compulsory.

1) The article could be improved by providing more details on the mechanisms and pathways involved in caffeine’s modulation of the intestinal microbiota and its metabolites, especially using figures to illustrate such mechanisms will greatly improve the manuscript.

Thank you for the comment. More details on the mechanisms of action and involved pathways have been added in the new version of the manuscript. Additionally, a new figure (Figure 2) has been included.

2) The authors could point out potential interactions and synergies with other dietary components or factors.

Thank you for the comment.  A nova versão do manuscrito contempla este aspecto quando existente. The new version of the manuscript addresses this aspect when applicable.

3) The article could also discuss the clinical implications and applications of caffeine’s anti-adipogenic effects for obesity prevention and treatment.

Thank you for the comment.  The item 6 "Impacts of caffeine on shaping obesity through intestinal microbiota and its metabolites" was rewritten to encompass this discussion (L538-L651).

Minor modifications needed, compulsory.

1)  Line 18. Space needed between the words “outcomes” and “In” in the phrase “outcomes.In”.

Thank you for the comment.   It was modified as requested.

2) Figure 1. Requires revision to enhance clarity. Resize the figure and improve the quality. What use is the big white arrow for, and why is it not mentioned in the body of the work? It will be good to commit to one type of arrow, unless the different arrow type is to typify a specific mechanism different from the general information been carried out.

Thank you for the comment. Figure 1 has been modified and referenced in the text as requested.

3) I prefer that a figure should be created to illustrate the caffeine mechanism of action on obesity. This will easily improve the manuscript and enhance the readership of this manuscript.

Thank you for the comment.  A new Figure 2 has been inserted into the text as requested.

Reviewer 2 Report

Comments and Suggestions for Authors

This review include a literature data, in vitro, in vivo, and pilot trials, about  the anti-adipogenic effect of caffeine in modulating metabolic pathways of obesity through the intestinal microbiota and its metabolites.

I have found the manuscript interesting and comprehensive with a lot of data that summarize the anti-adipogenic effect of caffeine  and may increase the knowledge in the field of the metabolites with healthy effects which can be explored in medicine and food industry.

The manuscript does not have data from the years 2022-2024 (only 3), so I suggest the authors to  improve the manuscript with more data from the last 2 years.

Upon a brief screening of the literature only in 2022-2024 there are many articles on this subject, the interaction of caffeine with microbiota and the involvement in obesity. At a glance, over 400 in 2022-2024.

I consider Table 1 very useful, synthetic, and can be completed with more data, if there is more.

Author Response

I hope this letter finds you well. I appreciate the time and effort that you and the reviewers have dedicated to the review of our manuscript titled “Metabolic Insights into Caffeine's Anti-adipogenic effects: An exploration through Intestinal Microbiota Modulation in Obesity." We are grateful for the insightful comments and suggestions provided by the referees. We have carefully considered each of the points raised, and I would like to address them in detail.

The manuscript does not have data from the years 2022-2024 (only 3), so I suggest the authors to  improve the manuscript with more data from the last 2 years.

Thank you for the comment. The text has been modified in its new version, updating the references

Upon a brief screening of the literature only in 2022-2024 there are many articles on this subject, the interaction of caffeine with microbiota and the involvement in obesity. At a glance, over 400 in 2022-2024.

In fact, when using the keyword 'caffeine' in the period 2022-2024, over 2000 articles appear. However, in the section 'Impacts of caffeine on shaping obesity through intestinal microbiota and metabolites,' aiming to correlate obesity, intestinal microbiota (with a focus on intestinal metabolites), and caffeine, only 10 articles were found through an advanced search. These 10 articles are already included in the updated version of the review.

The primary focus of this section was to describe the impact of caffeine on the development of obesity through both the modulation of the intestinal microbiota and intestinal metabolites. It's noteworthy that while the effect of caffeine on the modulation of obesity through the microbiota is well-documented, its impact on intestinal metabolites is less extensively described.

I consider Table 1 very useful, synthetic, and can be completed with more data, if there is more.

Thank you for the comment. Considering what was described above, the current version of Table 1 contains the available material up to the present date